The influence of temperature and photoperiod on the timing of brood onset in hibernating honey bee colonies

http://orcid.org/0000-0002-7360-3617 Nürnberger Fabian fabian.nuernberger@uni-wuerzburg.de
Härtel Stephan
Steffan-Dewenter Ingolf
Department of Animal Ecology and Tropical Biology, Bayerische Julius-Maximilians-Universität Würzburg , Würzburg , Germany
Paris Claire
Electronic publication date: 2018 May 25
Publication date: 2018
Volume: 6
Electronic Location ID: e4801
Received 2018 Jan 26; Accepted 2018 Apr 30
Copyright: © 2018 Nürnberger et al.
Copyright year: 2018
Copyright holder: Nürnberger et al.
License: This is an open access article distributed under the terms of the Creative Commons Attribution License, which permits unrestricted use, distribution, reproduction and adaptation in any medium and for any purpose provided that it is properly attributed. For attribution, the original author(s), title, publication source (PeerJ) and either DOI or URL of the article must be cited.
License URL: https://creativecommons.org/licenses/by/4.0/

Keywords: Phenology, Apis mellifera, Climate change, Winter cluster, Brood rearing activity, Thermoregulation

Funding: German Research Foundation (DFG) SFB 1047–Project C2 Funding was provided by the German Research Foundation (DFG) in the framework of the Collaborative Research Center 1047—Insect Timing: mechanisms, plasticity and fitness consequences, Project C2 (Ingolf Steffan-Dewenter and Stephan Härtel). The funders had no role in study design, data collection and analysis, decision to publish, or preparation of the manuscript.

==============================
In order to save resources, honey bee (Apis mellifera) colonies in the temperate zones stop brood rearing during winter. Brood rearing is resumed in late winter to build up a sufficient worker force that allows to exploit floral resources in upcoming spring. The timing of brood onset in hibernating colonies is crucial and a premature brood onset could lead to an early depletion of energy reservoirs. However, the mechanisms underlying the timing of brood onset and potential risks of mistiming in the course of ongoing climate change are not well understood. To assess the relative importance of ambient temperature and photoperiod as potential regulating factors for brood rearing activity in hibernating colonies, we overwintered 24 honey bee colonies within environmental chambers. The colonies were assigned to two different temperature treatments and three different photoperiod treatments to disentangle the individual and interacting effects of temperature and photoperiod. Tracking in-hive temperature as indicator for brood rearing activity revealed that increasing ambient temperature triggered brood onset. Under cold conditions, photoperiod alone did not affect brood onset, but the light regime altered the impact of higher ambient temperature on brood rearing activity. Further the number of brood rearing colonies increased with elapsed time which suggests the involvement of an internal clock. We conclude that timing of brood onset in late winter is mainly driven by temperature but modulated by photoperiod. Climate warming might change the interplay of these factors and result in mismatches of brood phenology and environmental conditions.

Introduction

The timing of life-history events, such as flowering in plants, insect emergence, and reproduction, with respect to the changing abiotic and biotic conditions of the environment is critical for most organisms (Van Asch & Visser, 2007; Visser, Both & Lambrechts, 2004). In temperate regions, environmental conditions during winter are important drivers of phenology (Williams, Henry & Sinclair, 2015) as organisms need to cope with low temperature conditions and often drastically reduced resource availability. Most ectotherms hibernate in a state of dormancy at different stages of development. Endothermic mammals generally keep their body temperature actively above ambient temperature, but often go into a state of reduced metabolism, i.e., hibernation or daily torpor, to reduce energy expenditure and tend not to reproduce during winter (Körtner & Geiser, 2000). Due to their capability of social thermoregulation, honey bees (Apis mellifera L.) are able to maintain colonies over the whole year (Jones & Oldroyd, 2006), using a strategy analogous to hibernation in mammals. Much like mammals that undergo hypothermic phases during hibernation, the honey bee colony is effectively heterothermic. When the colony experiences cold stress the workers of a colony tend to remain relatively inactive and cluster up densely in the so-called winter cluster to reduce colony heat loss (Southwick, 1985), while individual workers actively produce heat by flight muscle shivering to keep the cluster core temperature above ambient temperature (Esch, 1964; Stabentheiner, 2005). In brood rearing honey bee colonies, the degree and accuracy of thermoregulation is exceptionally high (Fahrenholz, Lamprecht & Schricker, 1989; Jones et al., 2004; Kronenberg & Heller, 1982). This is necessary as the larvae of honey bees require a higher and more stable temperature than workers to survive and develop well. Even minor deviations from the optimal temperature-window during development can lead to decreased fitness in adult workers (Jones et al., 2005; Tautz et al., 2003). Thermoregulation is highly energy demanding (Stabentheiner, Kovac & Brodschneider, 2010). To save resources while foraging is not possible, honey bee colonies refrain from large-scale brood rearing during temperate zone winters. Anticipating resource availability in spring, colonies resume brood rearing already in late winter. The timing of brood onset is critical for colony fitness (Seeley & Visscher, 1985). Premature brood onset increases the risk of starvation before spring bloom and can lead to increased loads of the brood parasite Varroa destructor. Late brood onset, on the other hand, decreases the ability to exploit spring bloom. In both ways, wrong timing of brood onset can result in reduced colony growth, colony reproduction, and increased mortality during hibernation. Emergence from hibernation before new resources are available is also seen in several mammal species. Increased risk of predation and starvation are hazarded in order to reproduce early so that the offspring has sufficient time to develop and build up resource storages or fat-tissue before the next winter (Körtner & Geiser, 2000; Meyer, Senulis & Reinartz, 2016).

To date, very little is known on how honey bee colonies achieve an optimal timing of brood onset and which environmental factors are used as predictive cues during winter. Across many taxa increasing ambient temperature and length of photoperiod serve as cues to time phenological events like emergence after hibernation or reproduction (Bradshaw & Holzapfel, 2007; Körtner & Geiser, 2000; Visser, 2013). In addition, endogenous circannual clocks can control the timing of hibernation (Körtner & Geiser, 2000). Nothing is known about the role of internal clocks for timing of brood onset in honey bees. But it is generally assumed that ambient temperature does affect brood rearing activity in honey bee colonies in winter and it has been shown that photoperiod can affect brood rearing activity in summer (Kefuss, 1978). Empirical evidence for effects of ambient temperature or photoperiod on brood rearing in winter, however, is still lacking. This is probably because tracking the status of brood rearing within the winter cluster is difficult and generally highly invasive. We argue that a new method to detect brood rearing without disrupting the winter cluster is necessary to increase our understanding of the phenology of brood rearing activity in honey bee colonies. In light of ongoing climate change, well-founded information on the impact of environmental conditions on honey bee phenology is critically needed if we want to assess potential consequences of climate change for one of the most ecologically and economically important pollinators (Potts et al., 2016). Climate change and especially changing winter conditions have already been shown to alter timing of life history-stages in many organisms (Williams, Henry & Sinclair, 2015) and resulting mismatches with the environment can lead to severe fitness losses in wild bees (Schenk, Krauss & Holzschuh, 2018).

In this study we demonstrated that tracking the daily temperature variation within the winter cluster allows to draw conclusions on the state of brood rearing in a minimally invasive way. We applied this method to investigate the effects of ambient temperature, photoperiod and elapsed time on the brood rearing status within the winter cluster of honey bee colonies. We expected ambient temperature to have a major effect on timing of brood onset that is modulated by photoperiod and elapsed time.

Materials and Methods

Study organism

Twenty-four equally sized colonies of A. mellifera carnica (Pollmann, 1879) headed by sister-queens were established in July 2014. Queens were artificially inseminated with 8–10 μl sperm of 10 drones all belonging to the same drone population in cooperation with the Institut für Bienenkunde, Oberursel, Frankfurt University. Artificial swarms with 600 g of workers and a queen were placed into two-storied miniPlus-hive boxes with 12 empty wax-sheet frames and fed with sugar syrup (Apiinvert; Südzucker, Mannheim, Germany) during August to October 2014 to enable comb construction and ensure sufficient honey stores. Colonies were treated against the brood parasite V. destructor using Bayvarol®-strips (Bayer AG, Leverkusen, Germany) for six weeks in August and September 2014 as a precaution measure. No visually noticeable signs of common diseases were detected during two-weekly colony monitoring until September 2014. It was confirmed that all colonies successfully reared worker brood before hibernation and all colonies were adjusted in September 2014 to make sure that they contained approximately the same amounts of workers and honey stores. All colonies were placed into two environmental chambers in December 2014 (12 colonies in each chamber) and kept at 0 °C daily mean temperature with daily oscillation from −3 °C during midnight to +3 °C at noon and under constant short-day conditions of 8 h photoperiod. Within the environmental chambers, each colony was connected to a separate flight arena with an individually controllable LED light source (36 cold white (6500 K) LEDs and six UV-LEDs; ∼2000 lx illuminance), diffused with a sandblasted glass cover (Fig. 1). Honey bees could enter the flight arena via a short tunnel. The tunnels were covered with reflective aluminium foil to increase the amount of light that passes from the arena into the hive box to be perceived by honey bees in the winter cluster. To identify effects of ambient temperature and photoperiod on brood rearing activity, individual temperature and light regimes were started at 28th January. All applicable institutional and national guidelines for the care and use of animals were followed.

Figure 1 Experimental hive setup.

(A) Honey bee colonies were placed into experimental hive boxes, based on two miniPlus-styrofoam boxes, each with six comb frames. Hive boxes were connected to a third styrofoam box that served as flight arena via a short tunnel. (B) An array of LEDs was installed into each flight arena and allowed to implement individual light regimes for each colony. (C) A thermo-sensor was installed into the wax of the second to fourth comb on both hive levels in a way that allowed to track temperature on both sides of the comb. (D) Within each hive level thermo-sensors on consecutive combs were installed in alternating order, either between the left and the middle third of the comb or between the right and the middle third of the comb. This pattern was reversed on the other hive level to maximize the area covered by thermo-sensors. This allowed to keep track of thermoregulatory activity within the experimental colonies at relative high spatial resolution without disturbing or disrupting winter clusters. A photoelectric barrier within the tunnel between hive box and arena connected to a data logger allowed to track honey bee traffic between hive box and arena. All colonies were placed into two dark environmental chambers. A wire mesh bottom in the flight arena and hive box and metal lid on the flight arena top facilitated temperature exchange through convection and conduction to make sure that the honey bee colonies were not isolated from ambient temperatures. Photo credit: Fabian Nürnberger.

Temperature regimes

To investigate the effects of ambient temperature on brood rearing in honey bee winter clusters, colonies were distributed equally into two temperature treatments (Fig. 2): In environmental chamber A the temperature remained at constant cold conditions of 0 ± 3 °C for 78 days after the start of the experiment as a control.

Imitating a spell of warm weather, ambient temperature in environmental chamber B was gradually upregulated to 11 ± 3 °C after day 30 and after a warm period of 15 days, ambient temperature dropped again to cold conditions of 0 ± 3 °C.

Figure 2 Temperature and light regimes.

At the start of the experiment, 24 honey bee colonies within experimental hive boxes were distributed equally among two environmental chambers (environmental chamber A and B) that differed in ambient temperature regime. Each colony was connected to its own flight arena with individually controllable light regime and distributed among three different light regimes (constant, increasing and peaking photoperiod) independently from ambient temperature regime. This allowed us to test for effects of ambient temperature and photoperiod in isolation as well as for interacting effects on brood rearing activity in honey bee winter clusters. Gray area: daily amplitude of ambient temperature.

At day 78 and day 75 respectively the experiment was terminated and all colonies were released from the environmental chambers and placed outside on a meadow at the campus of the University of Würzburg at 6th March 2015.

Light regimes

To check for effects of total photoperiod and photoperiod changes on brood rearing activity colonies were assigned to three different photoperiod regimes (Fig. 2): Constant photoperiod: short-day conditions with an 8 h light to 16 h dark cycle (8:16 LD), which reflects the minimum day length in Central Europe and served as control treatment.

Increasing photoperiod: steadily increasing duration of photoperiod, starting at 8:16 LD with daily increase of 2 min 40 s, which is a simplified but realistic scenario for Central Europe between winter and summer solstice.

Peaking photoperiod: photoperiod starting at 8:16 LD with a steady increase in photoperiod of 10 m 40 s each day for 45 days to a maximum of 16:8 LD, followed by a steady decrease of 10 min 40 s each day until the end of the experiment. This additional experimental scenario was introduced to allow examination of effects of photoperiod change independently from photoperiod duration.

Tracking of comb temperature

Comb temperature in each colony was tracked by eight thermo-sensors (Maxim Integrated DS1921G-F5 Thermochron iButton; 0.5 °C resolution) that were embedded into the central wax layer of combs to keep track of winter cluster activity (Fig. 1). Temperature was measured in 3 h intervals. At each interval, the sensor that measured the highest temperature was considered as being closest to the center of the winter cluster and used in the statistical analyses as measure for comb temperature. When the in-hive temperature was up regulated to over 30 °C and the daily variation was not higher than 1.5 °C, colonies were defined as brood rearing (Kronenberg & Heller, 1982). Ambient temperature for each colony was tracked via a thermo-sensor in the respective flight arena.

Statistics

The statistical software R version 3.4.0 (R Core Team, 2017) was used for data analysis. For each observation day colonies were classified as brood rearing if the comb temperature was stable with a daily amplitude of comb temperature ≤1.5 °C. A linear-mixed effects model was used to test for the effects of ambient temperature and comb temperature variability on mean comb temperature. Data was square root transformed to meet requirements of normal distribution. A contrast matrix was used post hoc to test for differences between individual factor levels. We used a generalized linear mixed-effects model for binomial data to test for interacting effects of temperature phase and light regime on the proportion of days during which brood rearing occurred in colonies for each temperature phase and light regime combination. Only data from environmental chamber B was used to analyse interactions between the environmental factors. Temperature in chamber A remained constant at all times, making its data inadequate to assess interactions. Differences between individual levels of factors were tested post hoc using Tukey’s test. The effect of photoperiod duration on the proportion of colonies that were rearing brood for each day was tested, using a generalized linear mixed-effects model for binomial data. A linear mixed-effects model was used to test for effects of direction of photoperiod change on probability of brood rearing. We used a linear mixed-effects model to test for the effect of the direction of change of photoperiod on the probability of brood rearing. Only data from colonies that were kept at constant low temperature conditions was used to test for effects of photoperiod duration or direction of change of photoperiod on brood rearing status. The effect of time spent within the experiment on proportion of colonies that reared brood was tested using a generalized linear-mixed effects model for binomial data. This was done for a subset of colonies under constant cold and short-day conditions, as well as for all colonies, regardless of treatment combination. Colony ID was included as random factor in all models. Benjamini–Hochberg correction for multiple testing was applied for all post hoc tests (Benjamini & Yekutieli, 2001). Model residuals were inspected visually to confirm normality and homoscedasticity. Sample sizes and degrees of freedom were based on numbers of observation days. For all models, a significance level (α) of 0.05 was considered.

One colony under constant cold temperature and peaking photoperiod conditions was removed from the statistical analyses because the temperature profiles revealed that it was still rearing brood at the beginning of the experiment and continued to rear brood during the whole experiment. Three colonies within environmental chamber A and one colony within environmental chamber B were removed from the analyses because they died early in the experiment. This left the treatment combination of constant cold temperature and increasing photoperiod with only two colonies. As data from all colonies within chamber A were combined to analyze effects of photoperiod, this should not have compromised statistical analysis. All other treatment combinations were left with at least three colonies. Four colonies were lost during the second half of the experiment. Observation days from these colonies were included into the analyses until temperature profiles became unstable and eventually dropped to ambient temperature level. A total of 1,325 observation days from 19 colonies contributed to the statistical analysis.

Results

Variability of comb temperature

Stability of comb temperature and mean ambient temperature had interacting effects on mean comb temperature measured in the winter cluster (interaction: stability of comb temperature × mean ambient temperature: F1, 1271.85 = 8.26; p = 0.004; n = 1,325 observation days from 19 colonies; Fig. 3). When comb temperature was stable (i.e., daily amplitude of comb temperature ≤1.5 °C) mean comb temperature was significantly higher than when comb temperature was variable (i.e., daily amplitude of comb temperature >1.5 °C; z = 6.19, p < 0.0001) and no longer affected by ambient temperature (Tukey’s post hoc test: z = 1.60, p = 0.111). This state of stable comb temperature was considered a strong indicator of brood rearing activity. Stable comb temperature was used to identify brood rearing activity in colonies for all following analyses. When comb temperature was variable, mean comb temperature was negatively correlated with ambient temperature (Tukey’s post hoc test: z = −3.35, p = 0.001). Colonies were considered to not rear significant amounts of brood in this state.

Figure 3 Decreased daily variation of comb temperature in honey bee colonies is accompanied by a significant increase of mean comb temperature which is no longer significantly affected by ambient temperature.

Linear mixed-effects model: stability of comb temperature × ambient temperature: F1, 1271.85 = 8.26; p = 0.004). Blue line: stable comb temperature, defined by daily amplitude of comb temperature ≤1.5 °C; black line: variable comb temperature, defined by daily amplitude of comb temperature >1.5 °C. Gray areas: 95% confidence intervals. n = 1,325 observation days from 19 colonies. Tukey’s test with Benjamini–Hochberg correction for post hoc analysis of effect of ambient temperature on mean comb temperature. **: p < 0.01; ns: p > 0.05.

Effects of ambient temperature and light regime on brood rearing activity

There was a significant interaction between the effects of ambient temperature and light regime on the proportion of days during which colonies reared brood (i.e., daily amplitude of comb temperature ≤1.5 °C; data from environmental chamber B; temperature conditions × light regime: F4, 34 = 2.26, p < 0.023; n = 752 observation days from 11 colonies; Fig. 4). Under short-day conditions, the probability of brood rearing increased when ambient temperature was increased (11 ± 3 °C; Tukey’s post hoc test: z = 4.34, p < 0.001). A drop of ambient temperature back to 0 ±3 °C after the warm period did not significantly reduce the brood rearing activity (Tukey’s post hoc test: z = −1.85, p = 0.146). Surprisingly, there was no significant effect of ambient temperature on brood rearing under conditions of increasing or peaking photoperiod.

Figure 4 Depending on the light regime, ambient temperature conditions affected the proportion of days during which colonies were found to rear brood.

Cold: colonies were kept at constant cold conditions for 30 days. Increase: a spell of warm ambient temperature for 15 days. Drop: after the phase of temperature increase, ambient temperature dropped again to cold conditions. Constant: constant short-day light regime with an 8 h photoperiod (n = 189 observation days from three colonies; blue); increasing: gradually increasing photoperiod starting at 8 h photoperiod (n = 284 observation days from four colonies; green); peaking: fast increase of photoperiod, starting at 8 h photoperiod, peaking at 16 h photoperiod and followed by a fast decrease of photoperiod at the same point of time when temperature dropped again (n = 279 observation days from four colonies; orange). See Fig. 2 for more information. Generalized linear mixed-effects model. Tukey’s test with Benjamini–Hochberg correction for post hoc analysis of differences between factors within light regimes. Letters: statistical groups. ns: p > 0.05. Mean ± SEM.

Under constant low temperature conditions of 0 ± 3 °C within environmental chamber A the duration of photoperiod had no significant effect on the proportion of colonies that reared brood (F1, 570 = 0.10, p = 0.755; n = 573 observation days from eight colonies; Fig. 5). The direction of change of photoperiod had no significant effect on the proportion of days during which colonies reared brood (F2, 8.09 = 1.72, p = 0.238; n = 573 observation days from eight colonies; Fig. 6).

Figure 5 The proportion of brood rearing honey bee colonies was not significantly correlated with the duration of photoperiod under constant cold conditions.

n = 573 observation days from eight colonies. Generalized linear mixed-effects model ns: p > 0.05. Gray area: 95% confidence interval.

Figure 6 The proportion of days during which colonies were found to rear brood was not significantly affected by the direction of change of photoperiod under constant cold conditions.

Mean ± SEM. Constant photoperiod: n = 225 observation days from three colonies; increasing photoperiod: n = 279 observation days from five colonies; decreasing photoperiod: n = 69 observation days from three colonies. Linear mixed-effects model. ns: p > 0.05.

Independent of the tested environmental factors, the proportion of colonies that reared brood (i.e., daily amplitude of comb temperature ≤1.5 °C) significantly increased over time in both a subset of colonies that were all kept at constant cold and short-day conditions without further environmental cues (F1, 222 = 3.81, p = 0.045; n = 225 observation days from three colonies; Fig. 7) as well as across all treatments (F1, 1320 = 24.47, p < 0.0001; n = 1,325 observation days from 19 colonies).

Figure 7 The proportion of brood rearing honey bee colonies increased significantly with the elapsed time.

Black: n = 1,325 observation days from 21 colonies regardless of temperature and light regime; blue: n = 225 observation days from three colonies under constant cold and short-day conditions. Generalized linear mixed-effects model. Gray areas: 95% confidence interval.

Discussion

We demonstrated that tracking comb temperature with thermo-sensors is a valuable minimally invasive method to track brood rearing activity in honey bee hives, even during winter. Applying this method, we could show that onset of brood rearing in honey bee winter clusters is affected by environmental conditions. In our experimental setting, colonies were more often found to rear brood after ambient temperature was increased than during the preceding cold period. Neither duration of photoperiod nor the direction of daily change of photoperiod alone had a significant effect on brood rearing activity within winter clusters. However, the light regime did affect the response of winter clusters to temperature changes. There was only a significant response to temperature increase in colonies that were kept at constant short-day. While interacting effects of different abiotic conditions could help to minimize the risk of premature brood onset, our results suggest that increasing winter temperatures and more frequent spells of warm weather due to global climate change could result in advanced timing of brood onset. This might cause mismatches with the environment with negative consequences for honey bee colony fitness and pollination services. Independent of the measured environmental factors, onset of brood rearing also became more probable with time, which could indicate the involvement of an internal clock.

This study is, to the best of our knowledge, the first where individual and combined effects of ambient temperature and photoperiod on honey bee winter cluster activity were investigated under controlled conditions. Our experimental design allowed us to keep track of honey bee colony thermoregulation and thereby brood rearing activity under defined environmental conditions and without disturbing the colonies. We provide an alternative approach to earlier studies which were either extremely invasive (Avitabile, 1978) or not conducted under winter conditions (Fluri & Bogdanov, 1987; Harris, 2009; Kefuss, 1978). Indirectly detecting brood rearing by tracking thermoregulatory activity via thermo-sensors within the comb wax allowed us to investigate honey bee colonies under winter conditions without severely affecting honey bee behavior and colony health. By analyzing patterns of daily comb temperature variation, we could identify days where colonies performed intensive thermoregulation. A daily comb temperature amplitude within the winter cluster of maximally 1.5 °C, despite a considerably higher ambient temperature amplitude, was accompanied by an increase of mean comb temperature to more than 30 °C. Further, in this state mean comb temperature was not affected by mean ambient temperature. Such conditions were previously measured in the presence of capped brood within the winter cluster (Kronenberg & Heller, 1982). When colonies rear brood, the cluster core temperature is highly important and needs to be stable to allow for a proper development of brood (Jones et al., 2005; Tautz et al., 2003). We conclude that daily temperature amplitude measured within the winter cluster is a good predictor for brood rearing activity. It is important to keep in mind that the spatial resolution of temperature data was limited and small brood nests might not have been detected in all cases. In fact, even in temperate zones continuous brood rearing during winter could be common, albeit at very limited extent (Avitabile, 1978; Harris, 2009; Szabo, 1993). Once the brood nest grows and colonies start to rear brood at considerable amounts, this can be expected to be reflected in the temperature data obtained from our experimental setting. Although some uncertainty about the status of the colony will remain, we argue that this indirect method is preferable over the much more invasive method of disrupting the cluster to visually assess brood status.

In our experiment brood rearing activity was rarely detected under cold environmental conditions (i.e., −3 to +3 °C). Once ambient temperature increased, colonies were more often found to rear brood. The effect of ambient temperature on brood rearing activity is not surprising. The energy demand of thermoregulation necessary for brood rearing increases with decreasing ambient temperature (Kronenberg & Heller, 1982). As the resources needed to fuel thermoregulation are strongly limited, honey bee colonies should refrain from brood rearing under cold environmental conditions (Seeley & Visscher, 1985; Southwick, 1991). With increasing ambient temperature thermoregulation, and hence brood rearing, becomes less cost intensive and more viable, even when colonies need to solely rely on storages. Ambient temperature was previously also shown to have a strong effect on timing of increased thermoregulation after hibernation in ants of the Formica-group (Rosengren et al., 1987) as well as timing of hibernation and emergence in mammals (Körtner & Geiser, 2000; Meyer, Senulis & Reinartz, 2016; Mrosovsky, 1990; Ruf et al., 1993). After colonies started to rear brood, a drop of ambient temperature did not immediately cause them to stop. Pheromones released by honey bee larvae are known to stimulate brood rearing and associated behaviors in workers (Pankiw et al., 2004; Sagili & Pankiw, 2009). Hence, the mere presence of brood might have stimulated the workers to continue brood care and keeping the brood combs warm, even when mean ambient temperature was as cold as 0 °C. This may cause honey storages to run out quickly and leave colonies starving. It is possible that, once triggered, only a disruption of honey or pollen stores will ultimately force a stop of brood rearing activity. It is important to note that, despite a relatively large increase of ambient temperature, the proportion of days during which we detected brood rearing activity in our experiment only increased by about 30%. This reaction was weaker than expected and suggests that further factors are involved in the timing of brood onset.

Our data revealed that photoperiod in isolation had no effect on brood rearing activity. Neither duration of photoperiod nor direction of change of photoperiod affected brood rearing under cold conditions. It might be possible that honey bees are not able to measure photoperiod when densely packed within the winter cluster. It has been suggested for mammals which hibernate in shelters and therefore have limited access to day light, that ambient temperature would be the most appropriate stimulus or zeitgeber for timing of emergence after hibernation (Davis, 1977; Körtner & Geiser, 2000; Michener, 1977; Mrosovsky, 1980; Murie & Harris, 1982). However, in our experiment light regime did alter the response of honey bee colonies during warmer conditions, when winter clusters were probably less dense and workers could leave the cluster. Adult emergence, reproduction and oviposition in the marine midge Clunio marinus is also known to be controlled by two environmental factors that need to occur in unison (Kaiser & Heckel, 2012). Increasing ambient temperature affected brood onset only at constant short-day conditions of 8 h photoperiod, but not in the other two light regimes in which photoperiod was considerably longer (about 12–18 h, depending on light regime) and increasing. These findings are not in line with suggestions that a short photoperiod elicits cannibalization of eggs and hence inhibits brood rearing activity (Cherednikov, 1967; Woyke, 1977). Several studies proposed that, irrespective of current duration of photoperiod, an increase in photoperiod has a positive effect on brood rearing activity while a decrease of photoperiod negatively affects brood rearing (Avitabile, 1978; Kefuss, 1978). The inhibitory effect of photoperiod treatments with increasing photoperiod on brood rearing under warm conditions in our study does not support these findings. However, most of the previous studies that investigated the effect of photoperiod on brood rearing activity either did not investigate brood rearing activity in winter (Fluri & Bogdanov, 1987; Kefuss, 1978) or did not control for other environmental conditions that might have affected brood rearing activity like ambient temperature (Avitabile, 1978; Fluri & Bogdanov, 1987). It was previously shown that brood rearing activity in colonies that were kept at constantly low mean ambient temperature of 6 °C were not affected by photoperiod (Harris, 2009). This is in line with our findings, that photoperiod matters only under warm conditions. Fluri & Bogdanov (1987) failed to find an effect of photoperiod under warm conditions, but investigated the effect of artificial shortening of the photoperiod in summer when colonies were already rearing large amounts of brood. Under these circumstances the effect of photoperiod might be reduced (but see Kefuss, 1978). Due to the experimental settings, we cannot disentangle if it was the longer duration of photoperiod or the fact that photoperiod increased that reduced brood rearing under warm conditions. It also remains to be investigated if a decrease of photoperiod during a warm period would affect brood rearing. Our results indicate that photoperiod was used as additional cue and might help to prevent premature brood onset due to spells of warm weather. However, according to our hypothesis, short photoperiod was expected to inhibit brood rearing while increasing photoperiod should have promoted brood rearing activity and not vice versa. This illustrates that further experiments on combined effects of temperature and photoperiod are needed. It is important to note that honey bees show considerable geographical variation with a number of subspecies and locally adapted ecotypes (Meixner et al., 2013). We used A. mellifera carnica as it is one of the most commonly used subspecies in central Europe and of high economical relevance. It is highly productive and expected to increase brood rearing activity relatively fast once conditions seem favorable. To which extent other subspecies and ecotypes might differ in their reaction to environmental cues remains to be investigated.

In addition to photoperiod and ambient temperature also elapsed time affected brood rearing in the honey bee colonies. Brood rearing activity was detected with increasing frequency over time and we observed brood rearing activity in one colony even at constant short-day and cold conditions. This suggests that colonies recommence brood rearing at some point regardless of environmental conditions. It has been shown for mammals that a circannual rhythmicity underlies the timing of hibernation and seasonal torpor, which can be entrained by photoperiod, ambient temperature and food-availability, but does not rely on these external zeitgebers (Collins & Cameron, 1984; Heldmaier & Steinlechner, 1981; Körtner & Geiser, 2000; Mrosovsky, 1986; Steinlechner, Heldmaier & Becker, 1983; Wang, 1988). Timing of honey bee brood rearing activity might also be controlled by an internal clock. The queen is the only individual of a colony that can live for several years and thus feature a true circannual clock. Previous work has shown that not only egg-laying activity but also the size of queen ovaries changes over the seasons, which might be controlled by an endogenous rhythm (Shehata, Townsend & Shuel, 1981). Potential changes in queen pheromone releases related to an increasing ovary size might then prime the colony’s workers for brood caring in late winter as queen pheromones are involved in the regulation of worker tasks (Slessor, Winston & Le Conte, 2005). Another reason for increased probability of brood onset over time might be the build-up of moisture within colonies. It was proposed that the humidity in colonies affects brood rearing activity and brood may serve to bind moisture generated by the metabolic activity of colonies which may otherwise be harmful (Omholt, 1987). Humidity within the different colonies might have varied and was not tracked during the experiment. The availability of resources might be another highly important factor for timing of brood rearing. Colonies that were supplemented with pollen in spring were previously found to start brood rearing earlier in the year (Mattila & Otis, 2006). It has also been shown that the nutritional status of individuals and food-availability can affect the response to environmental cues for timing of hibernation in mammals (Norquay & Willis, 2014; Ruf et al., 1993) and the thermoregulation in Formica-ants (Rosengren et al., 1987).

Conclusions

We conclude that brood rearing activity in hibernating honey bee colonies is highly sensitive to climatic conditions. Ambient temperature seems to be an important trigger for brood onset, but responses to temperature can be modulated by photoperiod. Climate change and associated more frequent warm weather events during winter (IPCC, 2014) have the potential to disrupt the synchronization between the seasonal timing of brood onset in honey bee colonies and flowering phenology. This can have profound negative consequences for colony fitness.

Supplemental Information

Supplemental Information 1 Data on daily measured in-hive temperature and environmental temperature.

Click here for additional data file.

We would like to thank Prof. Dr. Bernd Grünewald and Beate Springer from the Institut für Bienenkunde, Oberursel, Frankfurt University, for expert support and provision of inseminated sister-queens for the experiment. We thank Susanne Schiele for her excellent assistance with the beekeeping, Thomas Igerst and Norbert Schneider for assistance in the development and design of the experimental bee hive system as well as for technical support during the experiment and Dr. Conrad Wild for programming and supporting the control system.

Additional Information and Declarations

Competing Interests

Author Contributions

Data Availability

The authors declare that they have no competing interests.

Fabian Nürnberger conceived and designed the experiments, performed the experiments, analyzed the data, contributed reagents/materials/analysis tools, prepared figures and/or tables, authored or reviewed drafts of the paper, approved the final draft.

Stephan Härtel conceived and designed the experiments, contributed reagents/materials/analysis tools, authored or reviewed drafts of the paper, approved the final draft.

Ingolf Steffan-Dewenter conceived and designed the experiments, contributed reagents/materials/analysis tools, authored or reviewed drafts of the paper, approved the final draft.

The following information was supplied regarding data availability:

The raw data are provided in the Supplemental File.

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
