# Peer review of "The influence of temperature and photoperiod on the timing of brood onset in hibernating honey bee colonies"

_PeerJ, doi:10.7717/peerj.4801_

## Round 0.1 · original submission · Minor Revisions

Both reviewers found the manuscript interesting and valuable. They also both suggested revisions that you should carefully address in a revised version.

Reviewer 1 ·

Basic reporting

I find that this manuscript addresses an interesting question in a mostly clear way. I suggest that a bit more care needs to be taken with language throughout.

For example:
L 24 change "..sufficient worker force that allows to exploit floral resources in upcoming spring." to "...sufficient worker force that allows the exploitation of floral resources in the upcoming spring."
L32 change "Tracking in-hive temperature as indicator for.." to "Tracking in-hive temperature as an indicator of.."
L43 change ".. in respect to" to "..with respect to"
L87 change "In the light of.." to "In light of.."
L89 - 90 change "... for one of the ecologically and economically most..." to "...for one of the most ecologically and economically important..."
L95 change ".. minimal invasive.." to "..minimally invasive.."
L125 change ".. equally to two.." to "..equally into two.."
L273 change "..isolation.." to "..insulation.."

Experimental design

Please describe whether colonies were checked for signs of disease before experiments were started. The presence of parasites and disease can have an effect on behaviour.

Validity of the findings

The results of the study are interesting and intriguing. It is indeed curious that no significant effect of ambient temperature on brood rearing under conditions of increasing or peaking photoperiod was found. With this in mind, I find that the main findings are overstated at the beginning of the discussion and in the conclusion. In particular, concluding that ambient temperature played a "major" role could be toned down.

The method used for indirectly detecting brood rearing by tracking thermoregulation is interesting and will be useful in future studies. However I suggest that discussion of this method is overly long in the discussion section and can be reduced. Additionally, I recommend introducing the testing and validating of this method as one of the aims of the study to provide a better connection between the different sections of the manuscript.

Because the different honey bee subspecies, also referred to as "ecotypes" are thought to be locally adapted to different environments, it would perhaps be interesting to mention why you chose to test the carnica subspecies, and how other subspecies might be differentially affected by temperature and photoperiod.

Additionally, as you highlight the possible involvement of an internal clock on brood rearing, it would be interesting to provide a little more detail on how this could operate in honey bees.

Additional comments

This study examines the effect of temperature and photoperiod on brood rearing activity in honey bees. If the authors can address the comments I have made, I think this manuscript has the potential to make an interesting contribution to PeerJ, and can promote further study on the factors driving brood rearing in honey bees, and importantly how climate change might affect this activity and what the potential consequences might be.

Reviewer 2 ·

Basic reporting

The study by Nürnberger monitored the effects of ambient temperature, photoperiod and elapsed time on the brood rearing status within the winter cluster of honey bee colonies. This study is interesting and original in its goal to assess the effects of ambient temperature or photoperiod on brood rearing during winter in a minimal invasive way.

Experimental design

This paper is well suited to the journal and well written. In general, the methods are well described and suitable to perform the study.

Validity of the findings

The authors have discussed the experimental outcomes and generally dealt fairly with the idiosyncrasies of the results. For these reasons, I have only minor comments or suggestions.

Additional comments

Lines 135. It would be useful to have here a very brief description of what kind of scenario each light regime is trying to simulate. Otherwise they look a bit random.

Lines 233-234: It is confusing that you mention "the complete data set" but the result here refers to 21 colonies, and not the 24 experimental colonies you used. Please clarify.

Lines 294-295: Since "cold" is a subjective term, it would be useful to remind the reader here what you mean by cold. For example: "...brood rearing activity was rarely detected under cold conditions (i.e. 0-3 degrees).

Line 314: It would be interesting that the authors mention here any other factor that could be involved in the timing of brood onset (e.g. diet).

Line 333: Are you referring to an inhibitory effect of photoperiod increase or photoperiod decrease? Please clarify in the text.


Line 362: The role of chemical stimuli on the initiation of warming behaviour in the colony, and thus brood rearing activity, should also be considered here. Chemical communication whithin honeybee colonies is crucial for the onset of brood rearing and other associated behaviours in the colony, and this shouldn´t be omitted in the text.


Figure 4: Typo in the description of "peaking". Please write photoperiod instead of pgotoperiod

---

## Round 0.2 · accepted · Accept

You have well addressed the reviewers suggestions and made many improvements to the manuscript.

# Reviewer 2 ·

Basic reporting

No comment

Experimental design

No comment

Validity of the findings

No comment

Additional comments

The authors addressed all my suggestions, and in my opinion they made many improvements to the manuscript. I think the article meets now the PeerJ criteria and should be accepted as is.